# Assessment of Soil Erosion at Multiple Spatial Scales Following Land Use Changes in 1980–2017 in the Black Soil Region, (NE) China

**DOI:** 10.3390/ijerph17207378

**Published:** 2020-10-10

**Authors:** Haiyan Fang, Zemeng Fan

**Affiliations:** 1Key Laboratory of Water Cycle and Related Land Surface Processes, Institute of Geographic Sciences and Natural Resources Research, Chinese Academy of Sciences, Beijing 100101, China; fanzm@lreis.ac.cn; 2College of Resources and Environment, University of Chinese Academy of Sciences, Beijing 100049, China; 3State Key Laboratory of Resources and Environmental Information System, Chinese Academy of Sciences, Beijing 100101, China

**Keywords:** soil erosion, land cover change, Revised Universal Soil Loss Equation, topographic factor, northeastern China

## Abstract

Impact of land use and land cover change on soil erosion is still imperfectly understood, especially in northeastern China where severe soil erosion has occurred since the 1950s. It is important to identify temporal changes of soil erosion for the black soil region at different spatial scales. In the present study, potential soil erosion in northeastern China was estimated based on the Revised Universal Loss Equation by integrating satellite images, and the variability of soil erosion at different spatial scales following land use changes in 1980, 1990, 2000, 2010, and 2017 was analyzed. The regionally spatial patterns of soil loss coincided with the topography, rainfall erosivity, soil erodibility, and use patterns, and around 45% of soil loss came from arable land. Regionally, soil erosion rates increased from 1980 to 2010 and decreased from 2010 to 2017, ranging from 3.91 to 4.45 Mg ha^−1^ yr^−1^ with an average of 4.22 Mg ha^−1^ yr^−1^ in 1980–2017. Areas with a rate of soil erosion less than 1.41 Mg ha^−1^ yr^−1^ decreased from 1980 to 2010 and increased from 2010 to 2017, and the opposite changing patterns occurred in higher erosion classes. Arable land continuously increased at the expense of forest in the high-elevation and steep-slope areas from 1980 to 2010, and decreased from 2010 to 2017, resulting in increased areas with erosion rates higher than 7.05 Mg ha^−1^ yr^−1^. At a provincial scale, Liaoning Province experienced the highest soil erosion rate of 9.43 Mg ha^−1^ yr^−1^, followed by Jilin Province, the eastern Inner Mongolia Autonomous Region, and Heilongjiang Province. At a county scale, around 75% of the counties had a soil erosion rate higher than the tolerance level. The county numbers with higher erosion rate increased in 1980–2010 and decreased in 2010–2017, resulting from the sprawl and withdrawal of arable land.

## 1. Introduction

Soil determines the biological and geochemical cycles of the earth system and provides basic services such as food, clean water, and goods for humankind. Severe soil erosion can thus influence soil fertility, sustainable agriculture, and the functions of soil systems [1,2]. In recent decades, the rapid expansion of arable land at the expense of natural vegetation induces severe soil erosion and environmental problems such as land degradation, river and reservoir sedimentation, and freshwater pollution. A deep understanding of land use and land cover (LULC) change-induced spatiotemporal patterns of soil erosion is of great importance to land use management and sustainable development [2].

The impact of LULC change on soil erosion has been studied at multiple spatial scales, from plot [3,4], catchment [5], and regional [6], to global scales [7]. Borrelli et al. [7] concluded an that there was an overall increase in global soil erosion induced by cropland expansion, and 35.9 Pg yr^−1^ of soil was eroded from the earth surface. In Europe, soil loss from the agricultural land accounts for 93.5% of the total [7]. At a catchment scale, studies concluded that in comparison to climate change, soil erosion is more sensitive to LULC change [5,8]. At a plot scale, more attention was focused on the impact of key factors such as rainfall intensity, slope, vegetation type, and coverage on soil loss. For example, Anh et al. [3] examined the impact of interactions of vegetation cover and soil macronutrient levels on soil erosion at 30 plots in a hilly area, northern Vietnam. Anache et al. [4] found that soil loss from plots was affected by land use types and their combination patterns. Until today, more work has been done to quantify the impact of the changing LULC on soil erosion at an individual scale, whereas less attention was paid to identify their variations with different spatial scales following LULC changes.

Northeastern China (NEC) belongs to one of the three major black soil regions in the world, where the fertile soils have rich soil organic carbon and superior physical and chemical characteristics [9]. NEC is one of the most important grain production areas in China. However, the soil depth has decreased from around 100 cm in its original cultivation period to 20–30 cm since the 1950s [10], due to extensive expansion of arable land and increasing population pressure [11,12].

Since the end of the 20th century, the severe soil erosion has been recognized. In 2003, the State Development and Reform Commission and Ministry of Water Resources of China began a pilot project to control soil erosion in NEC, and some practices including interception ditch, farmland shelterbelt, contour tillage, terrace, check dams, etc. were used to control soil loss [11]. Additionally, many studies have been done in NEC using runoff plots [13,14], tracers [15], and soil erosion models [10,16]. The impacts of land use on soil erosion have also been studied in NEC. For example, Ouyang et al. [17] examined the combined impacts of land use and soil property changes in 1979–2014 on soil erosion in a 142.5 km^2^ Bawujiu Farm. Fang [18] examined the relative contributions of land use change and dam construction to sediment yield in the Shuangyang River catchment, NEC. At a large spatial scale, Jiang et al. [19] studied the impact of land use change in 2000–2010 on soil erosion in Heilongjiang Province. These studies improve our understanding of the impact of LULC change on soil erosion. However, the lack of a reliable estimate of the entire NEC forces the scientific community to resort to these pioneering studies. A regional and/or global study on soil loss is important to identify its changing patterns in space and to guide policies’ implementation. For example, variations of soil erosion in the world with changing land use were given by Borrelli et al. [7]. In the eastern Arc Mountains of Kenya, the assessment of the potential impacts of agricultural expansion allows a clear distinction of priority regions for soil control policies during the next 20 years [20]. However, these studies can just provide soil loss at one scale with LULC change, without considering soil loss at different scales.

Soil erosion modeling is an efficient method to explore LULC change’s impact on soil erosion [8]. The choice of the most suitable model is a logical process influenced by many factors, including land use, the characteristics of catchment, and the data available [21]. Physical and/or conceptual models such as the European Soil Erosion Model (EUROSEM) and the Water Erosion Project Prediction (WEPP) have superiority over the Revised Universal Soil Loss Equation (RUSLE) because they give attention to runoff in determining the erosion processes [22]. However, these models are complex and require substantial data, which usually preclude their broad application, especially in developing countries or remote regions. Empirical models such as the Soil Loss Estimation Model for Southern Africa (SLEMSA) are site specific, and the Modified USLE (MUSLE) requires monitored runoff and peak flow information [23]. In contrast, the Revised Universal Soil Loss Equation (RUSLE) is simple, flexible, and time and cost effective. Although the RUSLE model also has limitations in terms of reliability and its spatial coverage for large areas, it is more accurate and solves some of these problems when it is combined with geographic information system (GIS) and remote sensing (RS). Nowadays, it has been widely used at large scales such as at regional and/or global scales [7]. Furthermore, in the study area, it has been verified that it can yield a reasonably reliable soil loss estimation [24].

Therefore, although some work has been done in NEC using different methods, soil erosion characteristics of NEC with LULC changes are scarcely studied. The RUSLE model was used in the present study by integrating long-term RS images to (i) examine the temporal variations of soil erosion at regional, provincial, and county scales following LULC changes from 1980 to 2017, (ii) analyze the impacts of LULC changes on soil erosion during the past decades.

## 2. Materials and Methods

### 2.1. Study Area

NEC (38°42′–53°36″, 115°24′–135°12″) includes three provinces (i.e., Heilongjiang, Jilin, and Liaoning Provinces) and the eastern Inner Mongolia Autonomous Region (IMA) (Figure 1a). It covers an area of 1.24 million km^2^, with elevations ranging from −10 to 2666 m a.s.l. NEC is composed of three mountains, three plains, and three large rivers. The Da Hinggang Mountain, the Xiao Hinggang Mountain, and the Changbai Mountain lie in the northwest, northeast, and southwest, respectively. Three plains including Liao River Plain, Songnen Plain, and Sanjiang Plain are situated between the mountains. They were formed by the piedmont alluvial and flood sedimentation from Heilong River, Songhua River, and Tumen River that join at the east of Sanjiang Plain. There are 2300 rivers and many lakes scattered in the region. The total amount of water resource is around 199 billion m^3^, with 170.4 m^3^ on the ground and 68.1 billion m^3^ underground. Due to active industrial and agricultural activities as well as increasing population since the 1950s, water pollution and deficiency are severe [25]. The slopes range from zero in the plains to above 60° in the mountainous regions (Figure 1b).

The average annual total precipitation in 1970–2018 ranged from around 993 mm in the southeast to less than 242 mm in the northwest (Figure 1c), around 65% of which fell in summer (i.e., June–August). The mean monthly temperature varied greatly, with the highest mean monthly temperature (18 °C) in July, and the lowest temperature (−20 °C) in January.

Affected by the summer monsoon from the southeast and obstruction by the Chanbai Mountain, the vegetation in NEC is distributed zonally, from forest to the east of the Changbai Mountain, to a transition zone between forest and grasslands in the Songnen Plain, and to grassland to the west of the Da Hinggang Mountain (Figure 1a). Coniferous forest is mainly distributed in the mountainous areas, and broad-leaved forests are scattered in the plains. Arable land in the plains covers about one-third of the total area. The main crops are maize, soybean, and rice, occupying around 87% of the total. A single-tillage operation is typically used with a cultivation ploughing depth of around 0.25 m in late September and in early May. From October to April, arable land is left fallow with no vegetation cover due to cold weather.

The main soil types are Luvisols, Phaeozems, Chernozems, and Gleysols, occupying 37.5%, 23.3%, 7.4%, and 6.8% of the total area in NEC (Figure 1d). The parent materials are Quaternary lacustrine and fluvial sand beds or loess sediments [26]. Fuvisols are widely distributed in mountainous areas, while Phaeozems and Chaeozems, which are called black soils by the local people, mainly appear in the Songnen Plain [9]. The textural classes of the topsoil are silt–clay–loam to clay–loam (8–27% sand, 29–66% silt, and 26–40% clay), with soil organic carbon usually above 4%. The hilly regions with long slope in the mountain–plain transition zone are prone to soil erosion. Rill erosion dominates the study area. However, gullies also widely occur [16].

In the study area, agricultural lands occupy around 30% of the provincial area for Heilongjiang, Jilin, and Liaoning Provinces. The major crops are rice, corn, and soybean. In Heilongjiang Province, their area percentages are 0.26, 0.44, and 0.21. In Jilin and Liaoning Provinces, their area percentages are 0.13, 0.70, 0.03 and 0.11, 0.66, and 0.02, respectively (Table 1).

### 2.2. Data Collection

Several data layers are required to run the RUSLE model: a Digital Elevation Model (DEM), land use maps, a rainfall erosivity map, a soil erodibility map, and land use data.

The DEM was prepared from Shuttle Radar Topography Mission (SRTM) with 90 m resolution. It was provided by Geospatial Data Cloud site, Computer Network Information Center, Chinese Academy of Sciences [27].

The Harmonized World Soil Database (HWSD) Version 1.2 with 30 arcsecond resolution was obtained from Food and Agriculture Organization of the United Nations [28]. Soil properties such as percentages of sand, clay, and silt; depth of soil; and organic carbon are available from the HWSD.

The daily precipitation data in 1970–2018 at the 120 meteorological stations were acquired from National Climate Centre of the China Meteorological Administration [29].

The land use datasets with 30 m resolution were acquired from different sources. The datasets in 1980, 1990, and 2000 were obtained from the Resource and Environmental Sciences, Chinese Academy of Sciences (RES-CAS) with an overall accuracy of 92.9% [18], which has a hierarchical classification system of 25 land cover classes. The datasets in 2010 and 2017 were downloaded from the GlobeLand30 [30] and from the research team of Gong Peng in Tsinghua University [31]. The GlobeLand30 product was generated from Landsat TM/STE+ and the Chinese BJ-1 and HJ-1sensors. This product consists of nine land cover types including barren, grass, cultivated, forest, shrub, water, artificial, wetland, and ice. Its overall accuracy is around 80% in 2010 [32]. The land use data by Gong Peng has a similar land use classification system, including cropland, forest, grassland, shrubland, wetland, water, tundra, impervious, barren, and snow/ice, with an overall accuracy of 72.76% [33]. These datasets were reclassified into nine land use types (Table 2).

### 2.3. Description of the RUSLE and Data Treatment

The RUSLE was employed to estimate the soil erosion rate. The RUSLE is a linear equation used to estimate soil erosion by water from hillslopes. It is able to reduce a complex system to a quite simple one to predict long-term soil erosion rates over large areas while maintaining a through representation of the main factors that influence the process [7]. It combines five factors, including rainfall erosivity, soil erodibility, slope length and gradient, land use management, and soil conservation practices:
(1)A=R×K×LS×C×P
where A is the soil erosion rate (Mg ha^−1^ yr^−1^), R is the rainfall–runoff erosivity factor (MJ mm ha^−1^ h^−1^), K is the soil erodibility factor (t ha h ha^−1^ MJ^−1^ mm^−1^), LS is a combination of slope length L and slope gradient S factor (-), C is the crop/cover management factor (-), and P is the soil conservation factor (-).

The classic method to calculate R factor is to multiply rainfall energy and the maximum 30 min rainfall intensity for rainfall events [34]. However, such datasets are not available for many places in the world. In such instances, alternative methods have been proposed to calculate R factor using readily available precipitation data [35,36]. In the current study, the daily precipitation data in 1970–2018 were used to estimate mean annual R-factor values, using the method proposed by Zhang et al. [36]:(2)$$R_{i}=a\sum_{j=1}^{k}{\left(D_{j}\right)}^{b}$$
where *R*_i_ is the half-month R-factor (MJ mm ha^−1^ h^−1^ yr^−1^), and *D*_j_ is the erosive rainfall on day j. *D*_j_ is equal to actual rainfall when it is greater than 12 mm. Otherwise it equal to zero. Here, *a* and *b* are empirical parameters. A detailed description was given by Fang and Sun [10]. A co-kriging interpolation was used to derive an R-factor value map in ArcGIS 10.5 software (ESRI, Redlands, CA, USA). In NEC, the mean annual R-factor values ranged from 475 MJ mm ha^−1^ h^−1^ yr^−1^ in western NEC to 5320 MJ mm ha^−1^ h^−1^ yr^−1^ in southeastern NEC (Figure 2a).

Soil texture and soil organic matter of the top 30 cm of soil were available from the HWSD, and the K factor values were calculated by using the EPIC (Environmental Policy Integrated Climate) model method. This method has been successfully used in NEC [10,17]:
(3)$$K=0.2+0.3e^{\left[-0.0256SAN\left(1-SIL/100\right)\right]}*{\left(\frac{SIL}{CLA+SIL}\right)}^{0.3}*\left[1-\frac{0.25SC}{SC+e^{\left(3.72-2.95SC\right)}}\right]*[1-\frac{0.72SN}{SN+e^{\left(22.9SN-5.51\right)}}]$$
where *SAN* is the sand content (%), *SIL* is the silt content (%), *CLA* is the clay content (%), *SC* is the soil organic carbon content (%), and *SN* = 1 − *SAN*/100. The K factor values of the study region varied from 0 to 0.045 t ha^−1^ h MJ^−1^ mm^−1^ ha^−1^ (Figure 2b).

The L factor measures the slope length, and the S factor is proportional to the local slope. This study used the 2D approach to calculate the LS factor as proposed by Van Rompaey et al. [37], using the methods by McCool et al. [38] and Desmet and Govers [39]:(4)$$L_{\mathrm{ij}}=\frac{{\left(A_{i,j}+D^{2}\right)}^{m+1}-A_{i,j}^{m+1}}{D^{m+2}*x_{i,j}^{m}*22.13^{m}}$$
(5)$$x_{i,j}={\sin\alpha }_{i,j}+{\cos\alpha }_{i,j}$$
(6)$$m=\frac{\beta }{\left(1+\beta \right)}$$
(7)$$\beta =\frac{\left(\sin\theta /0.0896\right)}{\left[3*{\left(\sin\theta \right)}^{0.8}+0.56\right]}$$
(8)$$S=\left\{\begin{array}{c} 10.8\sin\theta +0.03,\theta <9\%\\ 16.8\sin\theta -0.5,\theta \geq 9\% \end{array}\right.$$
where L_i,j_ is the slope length factor of the grid cell with coordinates i and j, A_i,j_ is the contribution area at the inlet of a pixel (m^2^), D is the grid cell size (m); $\alpha_{i,j}$ is the slope aspect direction for a pixel; m is slope length exponent, β is empirical factor (-), and θ is the slope angle. The derived LS factor values ranged from 0.03 to 70.02. Higher values occurred in the mountainous regions and lower values in the plains (Figure 2c).

The *C* and *P* factors are closely linked to land use types [40]. In order to obtain reasonable C- and P-factor values, a detailed survey of the literatures published in NEC was first carried out [19,24,41,42,43]. Then, the C-factor values in forest, grassland, shrubland, cultivated land, and residential areas were assigned as 0.004, 0.043, 0.07, 0.228, and 0.03, and those in bare land, wetland and water bodies, and other unused lands were 1, 0, and 0.06, respectively. P-factor values in cultivated land and water body were set to 0.352 and 0, and in other lands they were set to 1 in the study area.

The estimated soil erosion rates were further classified into five erosion classes from low (0T), moderate (1–5T), high (5–10T), and very high (10–20T), to critical (>20T) where T represents soil loss tolerance of 1.41 Mg ha^−1^ yr^−1^ in NEC [6]. In order to verify the estimated results, some published results from runoff plots, natural slopes, and catchments were also used in the present study.

## 3. Results

### 3.1. LULC Changes in NEC

The land use types in 1980–2017 were shown in Figure 3 and numerically in Table 2. Spatially, forest was distributed in mountainous areas, and arable lands in the three plains (Figure 1). Among the land use classes, forest was predominant and covered 37% of the whole area, followed by arable land and grasslands covering 30% and 22%, respectively. These three kinds of lands occupied around 90% of the total. Shrubland, wetland, water body, and unused lands each covered 2% of the total area.

During the past decades, great changes in arable and forest lands occurred. Before 2010, arable land percentages increased from 27% in 1980, 29% in 1990, and 30% in 2000 to 35% in 2010, and then decreased to 32% in 2017, fluctuating from 41.34 million km^2^ in 1980 to 45.63 million km^2^ in 2010. In contrast, forest area percentages decreased from 38% in 1980, 36% in 1990, and 37% in 2000 to 34% in 2010, and then increased to 37% in 2017. The area percentages of grassland decreased too before 2000, ranging from 21% in 1980 and 20% in 1990 to 19% in 2000, and then increased sharply thereafter, occupying around 25% in 2010–2017. Shrubland percentages occupied around 4% before 2000 and less than 0.1% thereafter. The percentages of residential area in general increased from 1% in 1980 to 2% in 2017. On the contrary, the area percentages of wetland and water body decreased from 3% and 3% in 1980 to 0% and 1% in 2017, respectively. Unused lands kept unchanged through time. Since 2010, bare land has almost disappeared. Noticeably, great LULC change occurred in 2000–2010 (i.e., P3 in Table 2), except for those of the wetland and bare land in 2010–2017. Arable land also changed with time on different slopes. In areas with elevation above 200 m a.s.l and slopes above 5 degree, the distribution of arable land showed similar change patterns in different years, i.e., the areas of arable land increased from 1980 to 2010 and decreased in 2017. In contrast, in the low- elevation and gentle-slope regions, arable land area increased continuously (Figure 4c,d).

### 3.2. Changes in Soil Erosion

In 1980–2017, spatial patterns of annual soil erosion rates changed little (Figure 5). Annual soil erosion rates in NEC ranged from 0 in the plains to over 40 Mg ha^−1^ yr^−1^ in the mountainous areas, with an average of 4.22 Mg ha^−1^ yr^−1^ (Figure 6a). It was three times the soil loss tolerance (T). Annual soil loss reached around 5.24 million tons from NEC. Figure 6b indicates that 57% of NEC belonged to the low erosion class, followed by moderate erosion with area covering around 30%. The high, very high, and extreme erosion areas covered 6%, 3.5%, and 2.7% of the total, respectively. Land use greatly influences soil erosion. The rates of soil erosion on bare land, shrubland, and arable land were much higher than those on other land use. Soil loss from arable land changed from 40% in 1980 and 46% in 2010 to 34% in 2017 of the total in respective years with an average of 40% (Table 2).

The area percentages with different erosion classes differed with time (Figure 6). The area percentages in the low erosion class decreased from 59% in 1980 to 56% in 2010 and increased to 57% in 2017. Opposite changing patterns occurred for the area percentages with high, very high, and extreme erosion severities. The threshold values occurred in 2010. In contrast, the area percentages with moderate erosion severity increased continuously with time.

The rates of annual soil erosion presented an obvious spatial heterogeneity among the provinces and the IMA (Figure 6). The mean soil erosion rate was the highest in Liaoning Province (i.e., 9.43 Mg ha^−1^ yr^−1^) in 1980–2017, followed by Jilin Province, and the IMA. The lowest erosion rate occurred in Heilongjiang Province, with an average of 2.46 Mg ha^−1^ yr^−1^ (Figure 6c). Considering the area percentages, their annual soil losses accounted for 21%, 17%, 26%, and 35% of the total area in NEC, respectively.

Temporally, the soil erosion rates of the three provinces increased from 1980 to 2010 and then decreased from 2010 to 2017. The rates of soil erosion in Heilongjiang and Jilin provinces in 2017 were less than their counterparts in 1980. The annual erosion rate in the IMA fluctuated from 1980 to 2017 (Figure 6d).

Low and moderate erosion zones dominated the area, with similar temporal patterns to those in NEC (Figure 7). Specifically, the areas with low erosion class dominated Heilongjiang Province, Jilin Province, and the IMA, and the areas with moderate erosion class were dominant in Liaoning Province. In the three provinces, the area percentages with low erosion class decreased from 1980 to 2010 and increased from 2010 to 2017. In contrast, it continuously decreased with time in the IMA. The area percentages with moderate erosion class increased with time for Jilin Province and the IMA. In contrast, it increased in 1980–2010 and decreased from 2010 to 2017 for Heilongjiang Province, with the opposite changing trend in Liaoning Province.

The spatiotemporal characteristics of soil erosion severity were mapped for the 224 counties in NEC (Figure 8). In 1980–2017, the number of the counties with soil erosion rates higher than one T accounted for around 75% of the total. The counties with soil erosion rates higher than 5T and 10T accounted for 24% and 8% of the total, respectively (Figure 9). For the top 16 counties with erosion rate higher than 10T, 12 and 4 counties were distributed in the southeastern and southwestern NEC, respectively. Almost all the counties with soil erosion rate lower than one T were distributed in the plains (Figure 8).

Arable land continuously sprawled from the south to the north in 1980–2010, and a little withdrawal occurred in 2017. In 1980–2010, among the 224 counties, there were 52 counties in the low erosion zone (i.e., less than T), 119 counties with moderate erosion zones (1–5T), 37 counties with high erosion zones (5–10T), 14 and 2 counties with very high (10–20T) and extreme erosion (>20T) zones, respectively (Figure 9a). The number percentages of the counties with low, moderate, high, very high, and extreme zones occupied 23%, 53%, 16%, 6%, and 1%, respectively.

In 1980–2017, the counties with low erosion rate ranged from 20% to 25% of the total and remained unchanged after 2000. The counties in moderate soil erosion zone decreased from 54% in 1980 to 49% in 2017, among which the counties with soil erosion rate 1–2T decreased fastest, from 21% in 1980 to 17% in 2017. Noticeably, the numbers of counties with high and very high classes increased from 14% and 5% in 1980 to 18% and 8%, respectively (Figure 9b).

## 4. Discussion

### 4.1. Model Performance

In NEC, soil erosion rates have been identified using runoff plot [14], ^137^Cs tracer technique [15,44,45], and model estimation [10,42]. Comparisons of the RUSLE-derived rates of soil erosion in the present study to published results can evaluate the performance of the RUSLE.

At plot and slope scales, the RUSLE-derived rates of soil erosion in the current study were less than the reported results (Table 3). This difference could result from two aspects. One aspect is that the 90 m resolution topographic data can smooth slope gradient [6,46], resulting in lower soil erosion rates. The other aspect is the difference in time spans between the studies in literature and the current study. The rates of soil erosion have also been reported at larger scales, including at a 28.5 ha catchment in Heshan Farmland [15], at a 916 km^2^ Shuangyang river catchment [10], and in the Sanjiang Plain [42]. A comparison at a catchment scale is more meaningful since this study was done at a large spatial scale. Table 3 demonstrates that the RUSLE-derived rates of soil erosion in the current study are basically consistent with the published results.

### 4.2. Impact of LULC Change on Soil Erosion

The changes of soil erosion from 1980 to 2017 can be attributed to the LULC with time, due to population pressure and national agricultural policies [5,11,48]. Since the mid-20th century, extensive settlement and land exploitation began in NEC, and a large amount of woodlands were converted into arable land [11]. Because of reduced precipitation and more water use consumption by increased crop land, around 4% forest and 3% wetland were lost, and around 3381 km^2^ woodland was converted to arable land in 1990–2000 [12]. As a result, soil loss from arable land accounted for 42% of the total, followed by grassland, forest land, and other land uses (Table 4). NEC is the most important Chinese grain production base, and a series of policies further stimulated arable land reclamation in 2000–2010. For example, the rescission of the tax from agriculture in 2005 and the household responsibility system triggered more cultivated farmland at the expense of woodland (i.e., forest and shrublands), resulting in a 46% soil loss of the total from arable land in 2010 (Table 4). Comparable higher soil loss rates on arable land were also reported in other regions of the world. In the U.K., a total of 1566 individual records indicated that arable land had the highest mean erosion values among different land use types [49]. An average soil erosion rate of 4.04 Mg ha^−1^ yr^−1^ was also observed in the agricultural lands in the European territory of Russia [50]. For the part of Europe covered by the CORINE database, soil loss from arable land is 3.6 Mg ha^−1^, which is three times of the average rate [51]. Soil loss from the agricultural land in Europe accounts for 93.5% of the total in Europe. A review study by Guo et al. [52] also gave higher soil erosion rates of arable lands in Australia (11.0 Mg ha^−1^ yr^−1^), Africa (14.9 Mg ha^−1^ yr^−1^), and USA (18.378 Mg ha^−1^ yr^−1^). Higher soil loss rate on arable land in the study area and its area can explain that arable land was the major sediment source in NEC. To decrease soil loss, the “Grain to Green Program” (GTGP) was introduced. This program is the largest ecological restoration in the developing world, initiated from 1999 and has made a remarkable contribution to China’s vegetation recovery [53]. Similar programs also appeared in other countries. For example, the Great Green Wall of the Sahara and the Sahel (GGW), which covers 21 countries in Africa [54], the First Ten-year Forest Rehabilitation (TYFR) in South Korea, and the Five Million Hectare Reforestation Program (5MHRP) in Vietnam. In NEC, impacted by the GTGP, farmland shelterbelts and sand fixation forests were constructed, and conversion of farmland to forest occurred on steep slopes [11,12,36]. From 2010 to 2017, forest area increased by 39,928 km^2^, with the largest change ratio of 9% during 2010–2017 (Table 4). The changes in land use policies and associated arable land cultivation can explain the increased area percentages with low erosion class in 1980–2010 and the decreased percentages from 2010 to 2017. Impacted by the policies, continuous sprawl of arable land in 1980–2010 and its withdrawal in 2010–2017 occurred in the mountainous areas.

### 4.3. Impact of Topography on Soil Erosion

The spatial distribution of soil erosion rates and their changes with time can be explained by the topography in NEC. First, steep mountainous areas usually have higher LS values [55]. This distribution characteristics has been found in literature [1,56,57]. Second, more rainfall and higher rainfall erosivity occurred in the mountainous areas (Figure 2a) and lower values occurred in the plains (Figure 3). The spatial pattern of the R value in mountainous regions can also be found in Sudetes Mts., SW Poland [1]. In NEC, soil erosion rates increased with increasing slope gradients (Figure 4a,b). The rate of soil erosion above 300 m a.s.l. was four to five times that in areas below 200 m a.s.l. Due to steep slope and high R values, the rates of soil erosion were high in high-elevation and steep mountainous regions, and low in plains (Figure 4a,b). Therefore, the spatial distributions of soil erosion rates at regional and county scales were quite related to topographic factors (Figure 1 and Figure 6), which is consistent with the studies by Bakker et al. [58] and Latocha et al. [59]. This can explain the spatial distribution of soil erosion rates at regional and county scales in NEC (Figure 4 and Figure 6).

### 4.4. Study Limitations

The soil erosion rate was calculated through multiplying the five factors (i.e., R, K, LS, C, and P). The data accuracies, especially the values of the C and P factors, greatly influence the results. Without consideration, the varied C- and P-factor values with time could result in biased estimation [6]. Furthermore, the datasets with 90 m resolution could influence estimation accuracy at plot or slope scales. However, although finer resolution could improve estimation accuracy, higher computational capacity is required, and it could not change the variation patterns of potential soil erosion.

Except for data resolution, the preclusion of gully erosion from the RUSLE can also induce biased estimation. In NEC, gullies are widely distributed, and over 295,000 (ephemeral) gullies were found in this region [60]. The soil loss from gullies is high. In a reservoir catchment in Baiquan County, the sediment from gullies occupied around 40% of the total [61]. Similar result was also obtained by Huang et al. [62] in Heshan Farmland, NEC. At a catchment scale, models such as WaTEM/SEDEM (Water and Tillage Erosion Model/Sediment Delivery Model) [37] and TeTIS [16] have combined the soil erosion module and the sediment deposition module to simulate soil sediment yield. The combination of such modules with RUSLE could improve the estimation accuracy.

The uncertainty also comes from the applications of LS calculation equations and land use data sources. This kind of limitation has fully been explained by Alewell et al. [63], and thus detailed explanation was not given here. However, it should be stated that the application of the upslope area-based L factor calculation could improve the estimation accuracy. The differences in land use classification systems and data resolution from RES-CAS, GlobeLand30, and Gong Peng research group in Tsinghua University can also lead to some bias of the estimated soil erosion.

Uncertainty also came from the employed model in the present study. The origin version of RUSLE is based on a field plot. With increasing GIS-based computational technique, it can also be applied to regional and global scales, and some studies indicated there are substantial differences between the results from RUSLE and the measurements from microrelief [64,65]. In contrast, Borrelli et al. [7] also pointed out the difference is not substantial. Although the estimated soil erosion rates were potential and approximated ones, they can reflect the variations of soil erosion and its responses to LULC changes in the study region, the focus of the present study.

## 5. Conclusions

Assessment of the impact of regional LULC change on soil loss is of paramount importance to appropriately regulate land use. Soil loss characteristics and its changing patterns following LULC changes in 1980–2017 were explored in NEC using the RUSLE method at regional, provincial, and county scales, and some findings were obtained.

In NEC, higher rates of soil erosion occurred in the mountainous regions and lower values in the plains. In the study period, agricultural policies greatly influenced LULC that determine temporal changes in soil erosion rate at different scales. Soil erosion rates increased from 1980 to 2010 and decreased from 2010 to 2017. Spatially, the highest soil erosion rate of 9.43 Mg ha^−1^ yr^−1^ occurred in Liaoning Province and the lowest value in Heilongjiang Province. At a county scale, the counties in the southeast and southwest of NEC experienced the highest soil loss rate. In recent years, around 75% of the counties still suffered soil loss with soil erosion rates higher than the tolerance level. Arable land was the major sediment source, appropriate policies should be created to control soil loss through limiting the sprawl of arable lands in unfavorable regions, and land use management should also be done for the bare-, grass- and shrublands because these lands also suffered higher soil erosion rates. Because this study just described soil loss patterns at a large scale based on RUSLE, further study on the efficiencies of soil conservation practices at finer scales should be done in this area.

## Figures and Tables

**Figure 1 ijerph-17-07378-f001:**
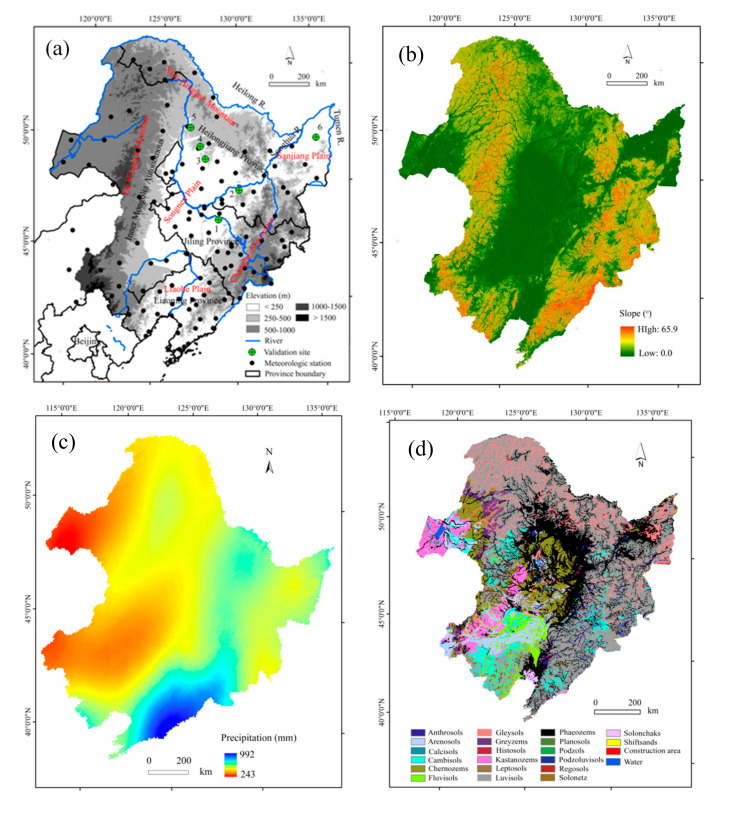
Study area showing the location, topography, meteorological stations, and the validation sites in the current study (**a**), slope gradient distribution (**b**), average annual total precipitation in 1970–2018 (**c**), and soil types (**d**). The numbers in Figure 1a indicate the sites of the published papers used for verifying our results in the current study (1 Dehui County; 2 Binxian County; 3 Baiquan County; 4 Keshan County; 5 Heshan Farmland; 6 Sanjiang Plain).

**Figure 2 ijerph-17-07378-f002:**
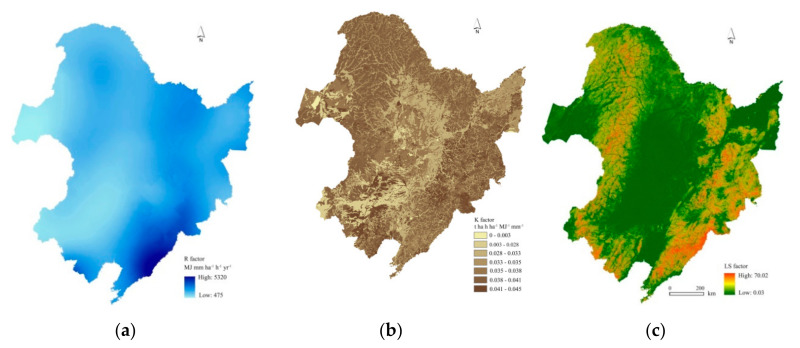
The Revised Universal Soil Loss Equation (RUSLE) input maps of the (**a**) R factor: rainfall–runoff erosivity factor (MJ mm ha^−1^ h^−1^), (**b**) K factor: the soil erodibility factor (t ha h ha^−1^ MJ^−1^ mm^−1^), and (**c**) LS factor: a combination of slope length L and slope gradient S factor (-).

**Figure 3 ijerph-17-07378-f003:**
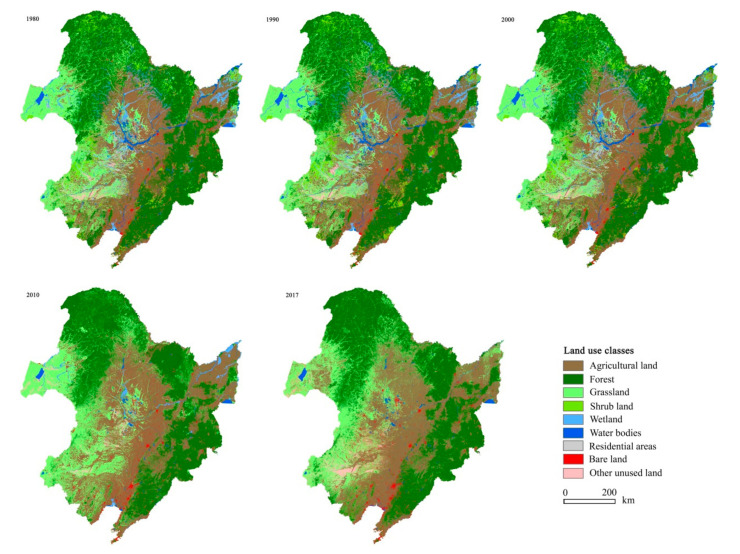
Land use maps in 1980, 1990, 2000, 2010, and 2017 for northeastern China (NEC).

**Figure 4 ijerph-17-07378-f004:**
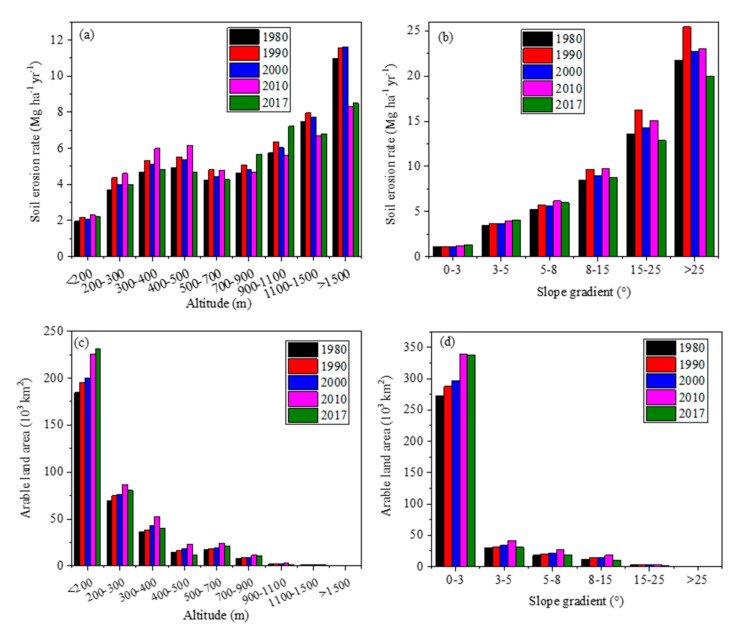
Distribution characteristics of soil erosion rates and the arable land areas according to altitude (**a**,**c**), and slope gradient (**b**,**d**) in the years 1980, 1990, 2000, 2010, and 2017.

**Figure 5 ijerph-17-07378-f005:**
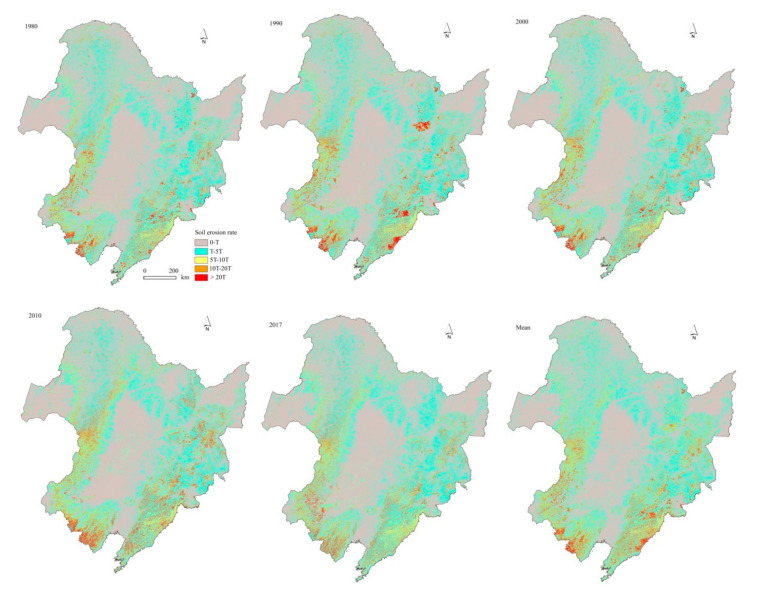
The spatial distribution characteristics of soil erosion rates in NEC in 1980, 1990, 2000, 2010, 2017, and their average in 1980–2017.

**Figure 6 ijerph-17-07378-f006:**
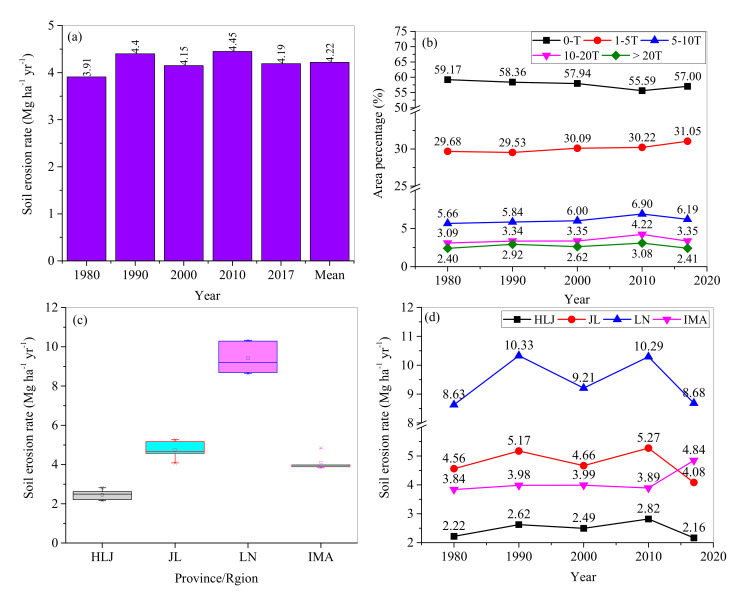
Changes of the mean annual erosion rate (**a**) and different erosion classes (**b**) in the whole of NEC with time, and soil erosion rates for three provinces and the Inner Mongolia Autonomous Region (IMA) (**c**) as well as their changes with time (**d**). The soil loss tolerance, T = 1.14 Mg ha^−1^ yr^−1^. HLJ, JL, LN, and IMA represent Heilongjiang Province, Jilin Province, Liaoning Province, and the eastern Inner Mongolia Autonomous Region.

**Figure 7 ijerph-17-07378-f007:**
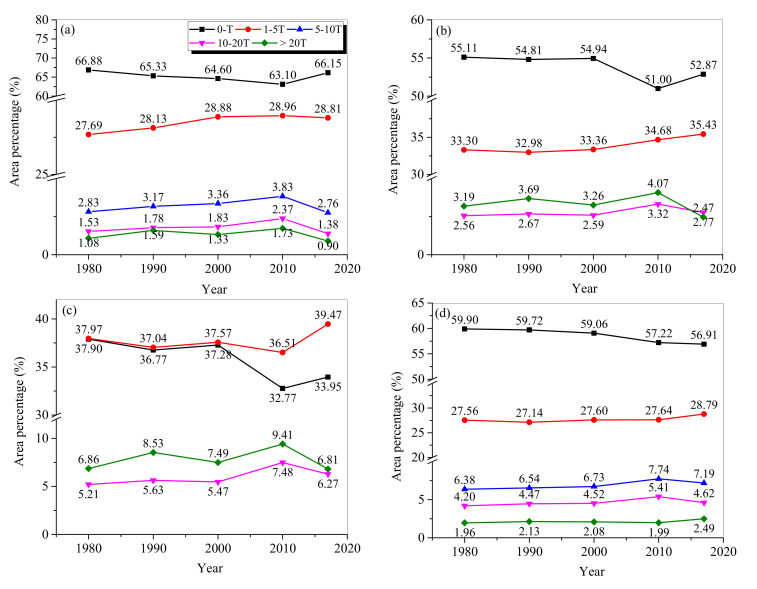
Area percentages of different erosion classes in Heilongjiang Province (**a**), Jilin Province (**b**), Liaoning Province (**c**), and (**d**) the IMA.

**Figure 8 ijerph-17-07378-f008:**
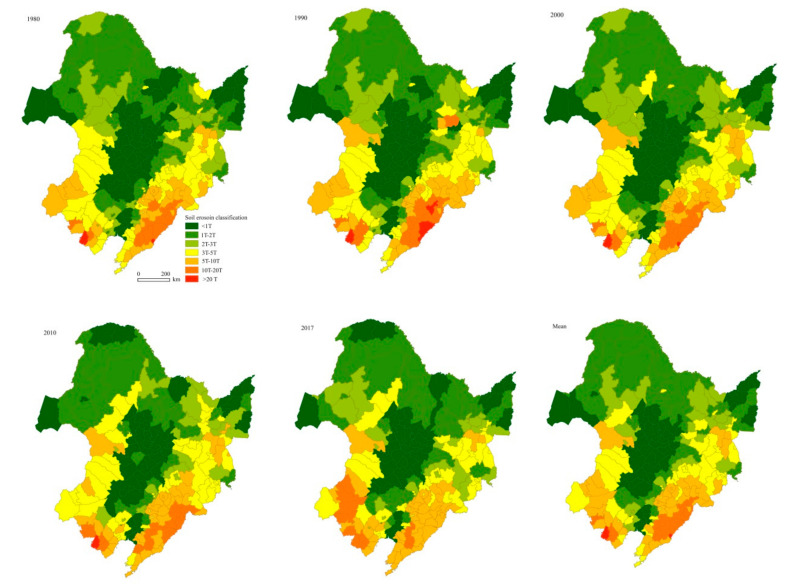
Maps showing the mean rate of soil erosion in 1980, 1990, 2000, 2010, 2017, and the average during the study period at a county scale in NEC.

**Figure 9 ijerph-17-07378-f009:**
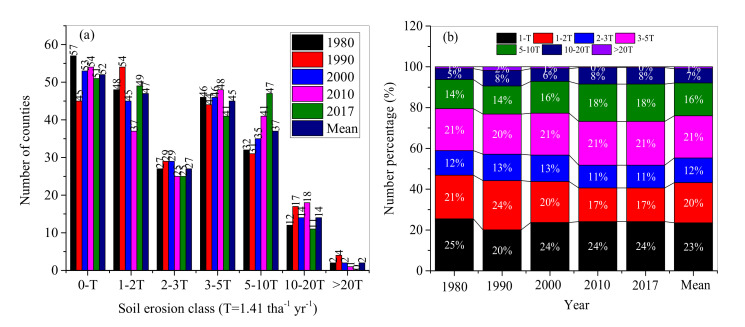
Changes of the number of counties (**a**) and the number percentages (**b**) with different soil erosion grades during 1980–2017.

**Table 1 ijerph-17-07378-t001:** The area (100 km^2^) and percentage (%) of the major crops in the three provinces. HLJ, JL, and LN represent Heilongjiang, Jilin, and Liaoning Provinces.

		Rice	Corn	Soybean	Wheat	Millet	Sorghum	Tuber	Others
HLJ	Area	381.0	644.3	312.5	11.3	3.2	6.1	20.8	93.6
	Percentage	0.26	0.44	0.21	0.01	0.00	0.00	0.01	0.06
JL	Area	80.0	424.2	18.8	-	-	10.6	6.1	66.6
	Percentage	0.13	0.70	0.03	-	-	0.02	0.01	0.11
LN	Area	47.6	279.0	7.0	0.3	3.8	3.7	7.9	75.0
	Percentage	0.11	0.66	0.02	0.00	0.01	0.01	0.02	0.18

**Table 2 ijerph-17-07378-t002:** Area percentage of each land use class in 1980–2017 and their change rates (R = 100 × (Area_i+1_ − Area_i_)/Area_i_; i represents the year used) in different time phases P1 (1980–1990), P2 (1990–2000), P3 (2000–2010), and P4 (2010–2017).

	%1980	%1990	%2000	%2010	%2017	%P1	%P2	%P3	%P4
Arable land	27	29	30	35	32	7	3	**16**	−7
Forest	38	37	37	34	37	−4	0	−7	**9**
Grassland	21	20	19	25	25	−4	−2	**30**	−3
Shrubland	3	4	4	0	0	20	−9	**−98**	67
Wetland	4	4	3	1	0	−5	−5	−66	**−98**
Water body	3	3	2	1	1	−1	−4	**−50**	0.00
Residential areas	2	2	2	2	2	4	−1	−1	**6**
Bare land	1	1	1	1	2	3	−11	36	**80**
Other unused lands	1	1	1	0	0	−19	24	**−100**	0

The bold value indicates the largest change rate for each land use and land cover (LULC) class in the row. Positive values indicate more land area occurred in the second year, and negative values indicate more lands in the first year.

**Table 3 ijerph-17-07378-t003:** The published rates of soil erosion in the black soil region (the study area locations of the references were labeled in Figure 1).

Sampling		Gradient (°)	Land Use	Size	Method	Erosion Rate (mm yr^−1^)	Erosion Rate * (Mg ha^−1^ yr^−1^)	Current Study (Mg ha^−1^ yr^−1^)	Reference
Region	Time								
1–1	2005	3–5	UPC	Natural slope	^137^Cs tracer	3.90	50.7	15.76	[14]
1–2	2005	2.2–3.5	UPC	Natural slope	^137^Cs tracer	1.87	24.31	12.81	[14]
3–1	2003–2004	6–10	UPC	Natural slope	^137^Cs tracer	-	29.0	20.0	[45]
5–1	2004	2.8	UPC	Natural slope	^137^Cs tracer	1.80	23.40	6.1	[14]
5–2	2005	0.5–3.5	UPC	Natural slope	^137^Cs tracer	1.00–2.66	21.76	11.4	[14]
4–1	1985–1990	5.0	Bare land	20 m plot	Monitoring	1.97	25.61	18.7	[14]
2	1985–1990	5.0	Bare land	20 m plot	Monitoring	3.31	43.03	16.4	[14]
5–1	2003–2004	5.0	Bare land	20 m plot	Monitoring	3.91	50.83	16.4	[14]
5–2	2003–2004	1.6	UPC	100 m plot	Monitoring	0.65	8.45	1.43	[18]
5–3	2003–2004	2.0	UPC	200 m plot	Monitoring	0.89	11.57	2.5	[14]
3–2	1980–1985	5	UPC	20 m plot	Monitoring	0.456	6.84	4.35	[47]
3–3	1980–1985	5	UPC	20 m plot	Monitoring	0.198	2.97	4.35	[47]
5–4	2011	0–3	UPC	28.5 ha catchment	^137^Cs tracer	-	2.2	2.8	[15]
5–5	2007	0–5	UPC	3 km^2^ catchment	^137^Cs tracer	-	11.10	2.67	[44]
3–4	2010	0–8	Forest	916 km^2^ catchment	Modeling	-	1.48	1.77	[10]
3–5	2010	0–8	Shrubland	916 km^2^ catchment	Modeling	-	2.58	5.27	[10]
3–6	2010	0–8.16	Grassland	916 km^2^ catchment	Modeling	-	1.09	4.33	[10]
3–7	2010	0–8.16	Multiple	916 km^2^ catchment	Modeling	-	2.58	2.54	[10]
6	2002	0–40	Multiple	108,900 km^2^	Modeling		2.26	1.81	[42]

The average bulk density 1.3 t m^−3^ was used when the unit mm yr^−1^ was converted to unit Mg ha^−1^ yr^−1^ according to Xie et al. [9]; CGWCR indicates the Complication Group of Water Conservancy Record in Baiquan County; UPC represents up–down cultivated land. The first numbers in the column “Region” represent the counties labeled in Figure 1. * indicates that the locations where the erosion rates were measured were given in Figure 1.

**Table 4 ijerph-17-07378-t004:** The mean soil erosion rate (Mg ha^−1^ yr^−1^), soil loss amount (SLA; 10^6^ Mg), and their percentage (%) of the total as well as their mean values of the five years for each land use class.

**LULC class**	**1980**			**1990**			**2000**		
	**Mean**	**SLA**	**Percent**	**Mean**	**SLA**	**Percent**	**Mean**	**SLA**	**Percent**
Arable	5.59	199.82	40	6.07	216.95	40	5.80	214.12	42
Forest	1.74	79.11	16	1.73	78.82	14	1.77	80.56	16
Grassland	4.05	99.74	20	4.30	105.86	20	4.34	104.28	20
Shrubland	15.83	79.05	16	18.37	91.70	17	15.27	69.33	13
Water area	0.00	0.00	0	0.00	0.00	0	0.00	0.00	0
Unused land	0.85	1.15	0	0.42	0.57	0	0.38	0.64	0
Residential	1.45	4.25	1	1.49	4.35	1	1.45	4.20	1
Bare land	34.49	42.19	8	39.03	47.74	9	37.95	41.50	8
	**2010**			**2017**			**Five-year mean**		
	**Mean**	**SLA**	**Percent**	**Mean**	**SLA**	**Percent**	**Mean**	**SLA**	**Percent**
Arable	5.98	256.52	46	4.47	178.53	34	5.58	213.19	40
Forest	1.76	74.57	14	1.77	82.13	16	1.75	79.04	15
Grassland	6.32	197.97	36	4.92	149.61	29	4.79	131.49	25
Shrubland	6.59	0.53	0	58.35	7.06	1	22.88	49.53	9
Water area	0.00	0.00	0	0.00	0.00	0	0.00	0.00	0
Unused land	9.58	0.01	0	8.71	0.00	0	3.99	0.47	0
Residential	1.31	3.76	1	1.88	5.72	1	1.52	4.46	1
Bare land	12.54	18.60	3	36.08	96.69	19	32.02	49.34	9

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
