# Peer review of "Assessment of Soil Erosion at Multiple Spatial Scales Following Land Use Changes in 1980–2017 in the Black Soil Region, (NE) China"

_ijerph, 2020, doi:10.3390/ijerph17207378_

Round 1
Reviewer 1 Report
Dear authors
I found your paper of quality and I would like to suggest some changes that can improve the final manuscript
Please, use Mg instead of t
The figures are of quality
The introduction needs to show why soil erosion is so important and why soil should be preserved
See my comments in the attached file
And more, the discussion is the only week point. I suggest you will discuss the important impact of the arable land in the final result of your study and compare to measurements (not USLE) done in other regions of the world to discuss why arable land is the source of most of the sediments
Sincerely

Author Response
Response to Reviewer 1 Comments Point 1: use Mg instead of t Response 1: Yes. It was replaced by Mg in the main manuscript, including that in figures. Point 2: The introduction needs to show why soil erosion is so important and why soil should be preserved Response 2: Yes, the significance of soil erosion and the causes to preserve it were added at the beginning of the introduction section. Simultaneously, the recommended references were added into the reference list. Point 3: And more, the discussion is the only week point. I suggest you will discuss the important impact of the arable land in the final result of your study and compare to measurements (not USLE) done in other regions of the world to discuss why arable land is the source of most of the sediments Response 3: Yes. More literatures were added, and some figures of soil loss rate on arable lands were cited to conclude that arable land was the major sediment source area in the study region. Some added references: Plambeck, N.O. Reassessment of the potential risk of soil erosion by water on agricultural land in Germany: Setting the stage for site-appropriate decision-making in soil and water resources management. Ecol. Indic. 2020, 118, 106732. Benaud, P.; Anderson, K.; Evan, M.; Farrow, L.; Glendell, M.; James, M.R.; Quine, T.A.; Quinton, J.N.; Rawlins, B.; Rickson, R.J.; Brazier, R.E. National-scale geodata describe widespread accelerated soil erosion. Geoderma 2020, 371, 114378. Guo, Q.K.; Hao, Y.F.; Liu, B.Y. Rates of soil erosion in China: A study based on runoff plot data. Catena 2015, 124, 68-76. Maltsev, K.; Yermolaev, O. Assessment of soil loss by water erosion in small river basins in Russia. Catena 2020, 195, 104726.Reviewer 2 Report
In this manuscript authors presented the research “Assessment of soil erosion at multiple spatial scales following land use changes in 1980-2017 in the black soil region, (NE) China”. The manuscript is well written and cover the all analysis part. However, manuscript need some revision in introduction, methodology, result, discussion section, need to correct table and figures, missing references to address the multiple issue for the publication in this journal.
Abstract: Add one sentence in abstract giving the information of land use and soil erosion and give the information of north-east-china.
Based on your Abstract and research work add one more keyword ‘Elevation’.
Introduction:
Please add some practice conducted by government to reduce the soil erosion and related policies in China, and if any others in NEC. Discuss more about global and regional level soil erosion and land use change issue. Please review other different models for soil erosion (USLE, MUSLE, SLEMSA, EUROSEM etc) and discuss more about RUSLE whish authors applied in this work. Add references in line number 54 after characteristics. Provide gap analysis in the end of introduction section.
Methodology: In this research work authors used DEM, Land use and soil data from different sources but authors missed to discussed more about the accuracy of these data base mainly Soil and land use data. Authors used Land cover data from different sources ( line 129), so need to discuss more about the accuracy of this data, Please add the accuracy of available database and compare the land cover result of both data for 2010 (RES-CAS and (http://www.globallandcover.com), if any different please mention in the limitation section. Furthermore. prepare a table for land cover classification scheme of presented data in result section.
Please check the font of equation-1-8.
Result section:
Compare the data of 2010 and 2010 from different sources or give details about the data source for 2010. I am not clear which data authors used in this table for 2010 ( it is from (RES-CAS and (http://www.globallandcover.com).
Line 238: according to the analysis it increased in 1980-2010 and decreased from 2010 to 2017. Please address this issue in discussion section including the major associated factors.
Figure 8a : make small font for this figure.
Discussion
Section 4.2 ( Line 290- 296) and 4.3 (Line 323- 333) insert this part in result section and describe more and other part of this section insert in discussion section.
Discuss more about different program like Grain to Green Program (GTGP) of China, Great Green Wall (GGW) for the Sahara. ecosystem restoration as part of the UN decade 2021-2031 as well as other work (from develop and developing country) to address the issue of your finding. Following work may support for review.
https://doi.org/10.1038/srep02846
https://doi.org/10.1016/j.ecoser.2019.100963
https://doi.org/10.1007/s10113-019-01545-0.
https://doi.org/10.3390/su11123488.
Conclusion
Re-write your conclusion section giving your finding of your research work, problem, solution, policy implementation and further way.
Author Response
Response to Reviewer 2 Comments
In this manuscript authors presented the research “Assessment of soil erosion at multiple spatial scales following land use changes in 1980-2017 in the black soil region, (NE) China”. The manuscript is well written and cover the all analysis part. However, manuscript need some revision in introduction, methodology, result, discussion section, need to correct table and figures, missing references to address the multiple issue for the publication in this journal.
Point 1: Abstract section, add one sentence in abstract giving the information of land use and soil erosion and give the information of north-east-china. Based on your Abstract and research work add one more keyword ‘Elevation’.
Response 1: Thanks. One sentence was added, and one key word of “elevation” was added. The added sentence is “Impact of land use and land cover change on soil erosion is still imperfectly understood, especially in northeastern China where severe soil erosion has occurred since the 1950s.”
Point 2: Introduction:
Please add some practice conducted by government to reduce the soil erosion and related policies in China, and if any others in NEC. Discuss more about global and regional level soil erosion and land use change issue. Please review other different models for soil erosion (USLE, MUSLE, SLEMSA, EUROSEM etc) and discuss more about RUSLE whish authors applied in this work. Add references in line number 54 after characteristics. Provide gap analysis in the end of introduction section.
Response 2: Yes. Soil control practices and related policies were added in the fourth paragraph of the instruction section. Global and regional level soil loss and LULC were discussed further. Other models like EUROSEM, WEPP, SLEMSA, MUSLE were discussed, and RUSLE was further discussed. Two references were added after characteristics. Gap analysis was given at the end of the introduction section.
Safwan, M.; Alaa, K.; Omran, A.; Quoc, B.P.; Nguyen, T.T.L.; Van, N.T.; Duong, T.A., Endre, H. Predicting soil erosion hazard in Lattakia Governorate (W Syria). Int. J. Sedi. Res. 2020, in press.
Kumar, P.S.; Praveen, T.V.; Prasad, M.A. Simulation of Sediment Yield over Un-gauged Stations Using MUSLE and Fuzzy Model. Aqut. Proc. 2015, 4, 1291-1298.
Ranzi, R.; Le, T.H.; Rulli, M.C. A RUSLE approach to model suspended sediment load in the Lo River (Vietnam): effects of reservoirs and land use changes. J. Hydrol.2012, 422-423, 17-29.
Point 3: Methodology: In this research work authors used DEM, Land use and soil data from different sources but authors missed to discussed more about the accuracy of these data base mainly Soil and land use data. Authors used Land cover data from different sources ( line 129), so need to discuss more about the accuracy of this data, Please add the accuracy of available database and compare the land cover result of both data for 2010 (RES-CAS and (http://www.globallandcover.com), if any different please mention in the limitation section. Furthermore. prepare a table for land cover classification scheme of presented data in result section.
Please check the font of equation-1-8.
Response 3: Very good suggestions. The accuracy of HWSD was added. The accuracies of the three classification systems were given. Sorry for the error, the year of 2010 should be 2000 for the datasets RES-CAS. Concrete land use types were also given. Because different land use systems, limitation was discussed and added in the discussion section.
The font of equations 1-8 were also checked and edited.
During the editions, some references were added:
Liu, J.Y.; Liu, M.L.; Tian, H.Q.; Zhuang, D.F.; Zhang, Z.X.; Zhang, W.; Tang, X.M.; Deng, X.Z. Spatial and temporal patterns of China’ cropland during 1990-2000: An analysis based on Landsat TM data. Remote Sens. Environ. 2005, 98, 442-456.
Hu, Q.; Xiang, M.T.; Chen, D.; Zhou, J.; Wu, W.B.; Song, Q. Global cropland intensification surpassed expansion between 2000 and 2010: A spatio-temporal analysis based on GlobeLand30. Sci. Tot. Environ. 2020, 746, 141035.
Gong, P.; Liu, H.; Zhang, M.N.; Li, C.C.; Wang, J.; Huang, H.B.; Cliton, N.; Ji, L.Y.; Li, W.Y.; Bai, Y.Q.; Chen, B.; Xu, B.; Zhuang, Z.L.; Yuan, C.; Suen, H.P.; Guo, J.; Xu, N.; Li, W.J.; Zhao, Y.Y.; Yang, J.; Yu, C.Q.; Wang, X.; Fu, H.H.; Yu, L.; Dronova, I.; Hui, F.M., Cheng, X.; Shi, X.L.; Xiao, F.J.; Liu, Q.F.; Song, L.C. Stable classification with limited sample: transferring a 30-m resolution sample set collected in 2015 to mapping 10-m resolution global land cover in 2017. Sci. Bull. 2019, 64, 370-373.
Point 4: Result section: Compare the data of 2010 and 2010 from different sources or give details about the data source for 2010. I am not clear which data authors used in this table for 2010 ( it is from (RES-CAS and (http://www.globallandcover.com).
Line 238: according to the analysis it increased in 1980-2010 and decreased from 2010 to 2017. Please address this issue in discussion section including the major associated factors.
Figure 8a : make small font for this figure.
Response: Truly sorry. The 2010 in RES-CAS should be 2000. It was corrected in the materials and methods section. Yes. It has been discussed in 4.2 section. To strength it, one sentence was added at the end of this section.
Figure 8a was edited according to your suggestion.
Point 5: Discussion
Section 4.2 ( Line 290- 296) and 4.3 (Line 323- 333) insert this part in result section and describe more and other part of this section insert in discussion section. Discuss more about different program like Grain to Green Program (GTGP) of China, Great Green Wall (GGW) for the Sahara. ecosystem restoration as part of the UN decade 2021-2031 as well as other work (from develop and developing country) to address the issue of your finding. Following work may support for review.
https://doi.org/10.1038/srep02846
https://doi.org/10.1016/j.ecoser.2019.100963
https://doi.org/10.1007/s10113-019-01545-0.
https://doi.org/10.3390/su11123488.
Response 5: Yes. The sentences in 4.2 (Line 290- 296) and 4.3 (Line 323- 333) were moved into result section, and some discussion were added. The important programs like GTGP, GGW and other vegetation restoration were added in section 4.3, and pointed out the vegetation restoration in the study area. Simultaneously, some references listed below were added.
Feng, X.M.; Fu, B.J.; Lu, N.; Zeng, Y.A.; Wu, B.F. How ecological restoration alters ecosystem services: an analysis of carbon sequestration in China’s Loess Plateau. Sci. Rep. 2013, 3, 2846.
Goffner, D.; Sinare, H.; Gordon, L.J. The Great Green Wall for the Sahara and the Sahel Initiative as an opportunity to enhance resilience in Sahelian landscapes and livelihoods. Reg. Environ. Change 2019, 19, 1417-1428.
Choi, G.; Jeong, Y.; Kim, S. Success Factors of National-Scale Forest Restorations in South Korea, Vietnam, and China. Sustainability 2019, 11, 3488.
Point 6: Conclusion
Re-write your conclusion section giving your finding of your research work, problem, solution, policy implementation and further way.
Response 6: The findings of this paper were summarized, problems were pointed and some ways were given to resolve these problems.
Reviewer 3 Report
This paper takes problem of the soil erosion estimation on large scale. The proposed methodology with the RUSLE approach is accepted but more criticized. The paper structure and text preparation are appropriate. However the some sections need correction and supplementation. Below I mention some problems with your article.
General comments:
- Abstract need correction according to journal guidelines, e.g. aims, methods, obtained results. You can order the following results from the regional scale to county scale.
- The Introduction section need correction. Please indicate more clarity reasons of your investigation and aims of this study.
- Materials and Methods section need supplementation. I propose to add one paragraph with agricultural condition, especially agrarian structure for study area and create new Table (the best as appendix) with land use percentage share for provinces and counties. You should descript more detailed the RUSLE modelling procedure with references.
- Discussion can be corrected. Your obtained results can be compare more carefully with selected subject literature.
More detailed comments are provided in the text (enclosed pdf).
To discussion:
- Problem with (extra)interpolation: Because the USLE model base on the plot scale data it is not correct in the methodology aspect use this approach to the (extra)interpolation to large scales e.g. catchment or regional scale. See the discussion in the denudation rate estimate of the empirical river sediment yield researches (e.g. Selby 1974; Jetten at al. 1999; Meybeck at al. 2003) or problem with unit erosion rate (e.g.: Poesen at al. 2003). Next the recent studies on the soil erosion mechanism in the subcatchment scale conducted by Janicki (2014, 2016) showed very big differentiation of really measurement soil erosion rate determined by: the microrelief, overland flow hydrodynamics and water drainage organization. Therefor you can add brief comments to quality of your erosion indexes and their only potential and approximation nature.
References:
Jetten V., De Roo A., Favis-Mortlock D., 1999. Evaluation of field-scale and catchment-scale soil erosion models. Catena 37 (3-4): 521–541.
Selby M.J, 1974. Rates of denudation. New Zealand Journal of Geography 56(1): 1–13
M Meybeck, L Laroche, HH Dürr, JPM Syvitski, 2003. Global and planetary change 39 (1-2), 65-93.
Poesen J., Nachtergaele J., Verstraeten G., Valentin C., 2003. Gully erosion and environmental change: importance and research needs. Catena 50 (2-4): 91–133.
Janicki G., 2014. Transformation of upland wash slope – a case study form the Lublin Upland (SE Poland). Annales UMCS, sec. B, 69(1): 31–48.
Janicki,2016 [https://www.researchgate.net/publication/324081063_System_stoku_zmywowego_i_jego_modelowanie_statystyczne_-_na_przykladzie_Wyzyn_Lubelsko-Wolynskich_in_Polish_with_summary_in_English_Wash_slope_system_and_their_statistical_modelling_-_case_study_from_]

Author Response
Response to Reviewer 3 Comments
This paper takes problem of the soil erosion estimation on large scale. The proposed methodology with the RUSLE approach is accepted but more criticized. The paper structure and text preparation are appropriate. However the some sections need correction and supplementation. Below I mention some problems with your article.
Point 1: Abstract need correction according to journal guidelines, e.g. aims, methods, obtained results. You can order the following results from the regional scale to county scale.
Response 1: Thanks. The abstract was edited according to your suggestions.
Point 2: The Introduction section need correction. Please indicate more clarity reasons of your investigation and aims of this study.
Response 2: Yes. The study aims were edited and added at the end of the manuscript. The reasons of the investigation were given at the end of paragraph 4 in the introduction section.
Point 3: Materials and Methods section need supplementation. I propose to add one paragraph with agricultural condition, especially agrarian structure for study area and create new Table (the best as appendix) with land use percentage share for provinces and counties. You should descript more detailed the RUSLE modelling procedure with references.
Response 3: We carefully checked the major crops in each province or county from the statistical yearbook in China, and found there are tens of crops. In the study area, there are 224 counties. Therefore, the table can be very large if the crops in each country were given. Therefore, the major crops were given for each province in the revised manuscript. More descriptions of the RUSLE model were given in the introduction and materials sections. Its limitation was also discussed in the 4.4 section. References were given when it was discussed.
Zerihun, M.; Mohammedyasin, M.S.; Sewnet, D.; Adem, A.A.; Lakew, M. Assessment of soil erosion using RUSLE, GIS and remote sensing in NW Ethiopia. Geoderma Reg. 2018, 12, 83-90.
Point 4: Discussion can be corrected. Your obtained results can be compare more carefully with selected subject literature.
Response 4: Yes, comparisons of the present results to other published literatures were done in the 4.2 section.
Point 5: To discussion, problem with (extra)interpolation: Because the USLE model base on the plot scale data it is not correct in the methodology aspect use this approach to the (extra)interpolation to large scales e.g. catchment or regional scale. See the discussion in the denudation rate estimate of the empirical river sediment yield researches (e.g. Selby 1974; Jetten at al. 1999; Meybeck at al. 2003) or problem with unit erosion rate (e.g.: Poesen at al. 2003). Next the recent studies on the soil erosion mechanism in the subcatchment scale conducted by Janicki (2014, 2016) showed very big differentiation of really measurement soil erosion rate determined by: the microrelief, overland flow hydrodynamics and water drainage organization. Therefore you can add brief comments to quality of your erosion indexes and their only potential and approximation nature.
Response 5 : Yes, a brief comments was given in 4.4 section, and some references were added. Furthermore, other suggestions in .pdf file were also edited.
References:
Jetten V., De Roo A., Favis-Mortlock D., 1999. Evaluation of field-scale and catchment-scale soil erosion models. Catena 37 (3-4): 521–541.
Selby M.J, 1974. Rates of denudation. New Zealand Journal of Geography 56(1): 1–13
M Meybeck, L Laroche, HH Dürr, JPM Syvitski, 2003. Global and planetary change 39 (1-2), 65-93.
Poesen J., Nachtergaele J., Verstraeten G., Valentin C., 2003. Gully erosion and environmental change: importance and research needs. Catena 50 (2-4): 91–133.
Janicki G., 2014. Transformation of upland wash slope – a case study form the Lublin Upland (SE Poland). Annales UMCS, sec. B, 69(1): 31–48.
Janicki,2016 [https://www.researchgate.net/publication/324081063_System_stoku_zmywowego_i_jego_modelowanie_statystyczne_-_na_przykladzie_Wyzyn_Lubelsko-Wolynskich_in_Polish_with_summary_in_English_Wash_slope_system_and_their_statistical_modelling_-_case_study_from_]
Reviewer 4 Report
The work is interesting and would be valuable for the specialist in this field, however couple of major questions need to be well justified along with the comments below, hope the comments would develop this research:
Abstract:
Please avoid using abbreviation in the abstract such as (LULC) , (NEC), (RUSLE),
Please remove this from the abstract (i.e., above 5 t ha-1 yr-1)
The author/s didn’t mentioned anything about the key of methods used in very short, the abstract usually is couple sentence for each ..Background, Hypothesis if any, key methods, key results. Please reflect this on your abstract.
Keywords
Please remove this abbreviation RUSLE or write what it stand for with no abbreviations.
Introduction
The objective of the study lines# 74-76 need more details and rephrase the paragraph to reflect an objective. Please provide more details.
- Materials and Methods
Please avoid having a title with no text, adding couple sentence before you move to another title or subtitle
The paragraph of the study area lines# 79-108 is providing very good information about the region of interest, however the paragraph is lack of the water resources status (recharge and discharge) for the region in both level surface water and groundwater, are the region relay on surface water or groundwater for agriculture? Is there and industrial activities that might lead to contamination or pollution? Please provide some details about the water resources, which will add significant value to the readers.
Line#116: (Revised Universal Soil Loss Equation) this is already identified in your previous paragraph there is no need to repeat what RUSLE is stand for. Please delate it.
Line# 122: (http://www.gscloud.cn). Please replace it with reference number and add the data link to the reference section. Same for Line# 124 (FAO-UN; http://www.fao.org) , line# 127: (http://data.cma.cn/).Line#131: (http://www.globallandcover.com) and line# 132: (http://data.ess.tsinghua.edu.cn/fromglc2017v1.html).
The author/s didn’t mentioned what is the novel of using the current methods and the RUSLE approach, a good example the two studies below what is the distinguished or add value for the current research in compare with the below two studies?
https://doi.org/10.1016/S2095-3119(18)62045-3
https://doi.org/10.1016/j.iswcr.2017.12.002
Please write a well justification for the current work add value in comparing with previous studies.
Results
Please avoid having a title with no text, adding couple sentence before you move to another title or subtitle
Lines# 197-198: “The area percentages of wetland and water body decreased from 3% and 197 3% in 1980 to zero and 1% in 2017, respectively” , Author/s didn’t given a reason for such a water degradation? Please justify the water availability declining and the shrinkage of the wetlands. Is this something to due to external challenge such as the climate change or internal challenges such as poor management or anything else? Same for the soil erosion in the next paragraph this also need to be justified.
A key question, do you think better DEM resolution might change your results significantly? If yes or not please add such a debate to your result or discussion paragraph because this would enrich the knowledge for the scholars later.
Author Response
Response to Reviewer 4 Comments
The work is interesting and would be valuable for the specialist in this field, however couple of major questions need to be well justified along with the comments below, hope the comments would develop this research:
Abstract section:
Point 1: Please avoid using abbreviation in the abstract such as (LULC), (NEC), (RUSLE), please remove this from the abstract (i.e., above 5 t ha-1 yr-1)
Response 1: It has been done.
Point 2: The author/s didn’t mentioned anything about the key of methods used in very short, the abstract usually is couple sentence for each ..Background, Hypothesis if any, key methods, key results. Please reflect this on your abstract.
Response 2: It has been done according the suggestion.
Point 3: Please remove this abbreviation RUSLE or write what it stand for with no abbreviations.
Response 3: “RULSE” has been replaced by its full name in the revised manuscript.
Introduction section:
Point 1: The objective of the study lines# 74-76 need more details and rephrase the paragraph to reflect an objective. Please provide more details.
Response 1: Thanks. It has been rewritten.
Materials and Methods section:
Point 1: Please avoid having a title with no text, adding couple sentence before you move to another title or subtitle.
Response 1: For this suggestion, we checked some published papers, and found it is not required to add a couple of sentences between two (sub) titles.
Point 2: The paragraph of the study area lines# 79-108 is providing very good information about the region of interest, however the paragraph is lack of the water resources status (recharge and discharge) for the region in both level surface water and groundwater, are the region relay on surface water or groundwater for agriculture? Is there and industrial activities that might lead to contamination or pollution? Please provide some details about the water resources, which will add significant value to the readers.
Response 2: Yes, more information could attraction more scientists. Information on water resource and water pollution in the study region has been added in this section.
Point 3: Line#116: (Revised Universal Soil Loss Equation) this is already identified in your previous paragraph there is no need to repeat what RUSLE is stand for. Please delete it.
Response 3: Yes. It was done.
Point 4: Line# 122: (http://www.gscloud.cn). Please replace it with reference number and add the data link to the reference section. Same for Line# 124 (FAO-UN; http://www.fao.org) , line# 127: (http://data.cma.cn/).Line#131: (http://www.globallandcover.com) and line# 132: (http://data.ess.tsinghua.edu.cn/fromglc2017v1.html).
Response 4: It has been done.
Point 5: The author/s didn’t mentioned what is the novel of using the current methods and the RUSLE approach, a good example the two studies below what is the distinguished or add value for the current research in compare with the below two studies? Please write a well justification for the current work add value in comparing with previous studies.
https://doi.org/10.1016/S2095-3119(18)62045-3
https://doi.org/10.1016/j.iswcr.2017.12.002
Response 5: Thanks. According to the suggestion and the reference given, the novelty was given at the end of introduction section and in the abstract section.
Results section:
Point 1: Please avoid having a title with no text, adding couple sentence before you move to another title or subtitle
Response 1: For this suggestion, we checked some published papers, and found it is not required to add a couple of sentences between two (sub) titles.
Point 2: Lines# 197-198: “The area percentages of wetland and water body decreased from 3% and 197 3% in 1980 to zero and 1% in 2017, respectively” , Author/s didn’t given a reason for such a water degradation? Please justify the water availability declining and the shrinkage of the wetlands. Is this something to due to external challenge such as the climate change or internal challenges such as poor management or anything else? Same for the soil erosion in the next paragraph this also need to be justified.
Response 2: The causes (natural and anthropogenic activities) inducing withdrawal of wetland and waster body as well as changed soil erosion were added in the 4.2 section.
Point 3: A key question, do you think better DEM resolution might change your results significantly? If yes or not please add such a debate to your result or discussion paragraph because this would enrich the knowledge for the scholars later.
Response 3: Different resolutions of DEMs could influence estimated results. This kind of study has been done. However, I don’t think it could greatly influence results because the study area covers an area of 1.24 million km2, the current 90 m resolution DEM is ok. Finer DEM resolution could not change the changing patterns at spatial and temporal scales impacted by decades of land use.
Round 2
Reviewer 2 Report
Dear authors thank you very much for your revised version of the manuscript.
Now need to correct all syntax error, missing reference,spelling error check your table number and figure and mention in main text also. Check reference number 37 why there is yellow colour. Province /Rgion or Region, correct it ( check the caption of figure 5.
Author Response
Point 1: Now need to correct all syntax error, missing reference, spelling error check your table number and figure and mention in main text also. Check reference number 37 why there is yellow color. Province /Region or Region, correct it ( check the caption of figure 5.
Response 1: The manuscript was carefully read and checked, and missing references (references 64 and 65), spelling errors, Table number, Figure number were edited. Reference number 37 was correct, and the yellow color was changed. For figure 5, the errors were corrected. Furthermore, we carefully read the manuscript, and corrected other errors too.
Reviewer 3 Report
Some comments to the revision version:
Line 33: Keywords - I propose change: 'elevation' on 'topography factor'.
Line 346: Table - Add unit to the Current study.
Author Response
Point 1: Line 33: Keywords - I propose change: 'elevation' on 'topography factor'.
Response: Yes, it was replaced.
Point 2: Line 346: Table - Add unit to the Current study.
Response: Yes, the unit was added.
Reviewer 4 Report
Thanks for the authors to consider all the comments and revised the work significantly, the work in current status looks much better wishing the authors all the best.
Author Response
Thanks for your comments. The manuscript has been carefully checked to correct some small errors.